# Microstructures and Mechanical Properties of Precipitation-Hardenable Magnesium–Silver–Calcium Alloy Sheets

**Mingzhe Bian \*, Xinsheng Huang and Yasumasa Chino**

Multi-Material Research Institute, National Institute of Advanced Industrial Science and Technology (AIST), Nagoya, Aichi 463-8560, Japan; huang-xs@aist.go.jp (X.H.); y-chino@aist.go.jp (Y.C.)
\* Correspondence: mingzhe.bian@aist.go.jp

**Abstract:** Precipitation hardening provides one of the most common strengthening mechanisms for magnesium (Mg) alloys. Here, we report a new precipitation-hardenable Mg sheet alloy based on the magnesium–silver–calcium system. In a solution treated condition (T4), the strength of Mg–xAg–0.1Ca alloys is enhanced with increasing the Ag content from 1.5 wt.% to 12 wt.%. The Mg–12Ag–0.1Ca (wt.%) alloy sheet shows moderate tensile yield strengths of 193 MPa, 130 MPa, 117 MPa along the rolling direction (RD), 45° and transverse direction (TD) in the T4-treated condition. Subsequent artificial aging at 170 °C for 336 h (T6) increases the tensile yield strengths to 236 MPa, 163 MPa and 143 MPa along the RD, 45° and TD, respectively. This improvement in the tensile yield strength by the T6 treatment can be ascribed to the formation of $AgMg_4$ precipitates lying on the $\{11\bar{2}0\}_\alpha$ and pyramidal planes. Our finding is expected to stimulate the development of precipitation-hardenable Mg–Ag-based wrought alloys with high strength.

**Keywords:** magnesium alloys; rolling; strength; segregation; precipitate

## 1. Introduction

The lightest structural metal magnesium (Mg) and its alloys have attracted significant interest in the past two decades due to their potential applications in the automotive sector [1–3]. However, current applications of wrought Mg alloys as a structural component are very limited, which can be ascribed to their inferior mechanical properties at room temperature (RT) and poor corrosion resistance [4]. Precipitation-hardening, also known as age-hardening, is one of the most effective ways to strengthen Mg alloys [5,6]. For example, an extraordinary high tensile yield strength of 473 MPa was obtained in a heavy rare-earth (RE) containing Mg–10Gd–5.7Y–1.6Zn–0.7Zr (wt.%) alloy processed by a conventional extrusion process followed by 200 °C aging for 64 h [7]. A RE-free Mg–6.2Zn–0.4Ag–0.2Ca–0.4Zr (wt.%) alloy subjected to twin-roll cast (TRC), hot-rolling and artificial aging at 160 °C for 24 h also yielded a high tensile yield strength of 316 MPa with a large fracture elongation of 17% [8]. Recent studies reported that Mg–Al–Ca–Mn(–Zn) dilute alloys could reach their peak hardness condition at 200 °C within 1 h that is significantly shorter than the time to reach the peak-hardness conditions of concentrated Mg–Gd, Mg–Zn and Mg–Sn based alloys [9–11]. Microstructure characterization by transmission electron microscopy (TEM), in conjunction with atom probe tomography, revealed that mono-atomic layer Guinier–Preston (G.P.) zones lying on the (0002) basal plane are formed in the peak-aged samples, and their dense distribution is responsible for the strength improvement in Mg–Al–Ca–Mn(–Zn) alloy sheets [12]. A more recent study demonstrated that bake-hardenability is even obtainable after 170 °C aging for only 20 min in a TRC Mg–1.3Al–0.8Zn–0.7Mn–0.5Ca (wt.%) alloy sheet, which opened up a possibility to develop bake-hardenable Mg alloys [13]. Those studies suggest that the development of precipitation-hardenable alloys is a promising approach to strengthen wrought Mg alloys.

In a recent study, we found that the co-addition of Ag (1.5 wt.%) and Ca (0.1 wt.%) to pure Mg could significantly enhance the stretch formability compared to the single addition of Ag or Ca [14]. A subsequent study achieved a high tensile yield strength of 182 MPa along the rolling direction (RD) by increasing the Ag content to 6 wt.%, while keeping a high index Erichsen value of 8.7 mm [15]. In a more recent study, we found that a compositionally optimized Mg–1.5Ag–2Ca (wt.%) alloy is non-flammable up to 1000 °C due to the formation of compact and dense CaO film on the surface. [16]. If precipitation-hardenability can be obtained by artificial aging, the Mg–Ag–Ca system is expected to attract more attention. Based on the binary Mg–Ag phase diagram, the maximum solid solubility of Ag is about 15 wt% in Mg at 472 °C, and it decreases to about 0 wt.% at 200 °C [17]. As shown in Figure 1, the equilibrium mole fraction of AgMg$_4$ phase is about 0.19 at 200 °C for the binary Mg–15Ag (wt.%) alloy, indicating there is a high possibility to obtain precipitation-hardenability from the Mg–Ag system. In fact, the Ag addition was reported to enhance the age-hardening response and thus increase mechanical properties of Mg–Nd alloys more than 50 years ago, which has facilitated the development of a commercial casting alloy QE22 (Mg–2.5Ag–2Nd–0.7Zr in wt.%) [18]. Subsequent studies showed that the Ag addition is also very effective in improving the age-hardening response of Mg–6Gd–0.6Zr and Mg–6Y–1Zn–0.6Zr (all in wt.%) casting alloys [19,20]. TEM observations revealed that the remarkable improvement in age-hardening could be ascribed to a dense and uniform distribution of nano-scale basal precipitates. On the other hand, adding a trace amount of Ag (0.4 wt.%) to a binary Mg–6.2Zn alloy was reported to enhance the age hardening response significantly due to a substantial refinement of MgZn$_2$ precipitates [21]. However, a closer look at those precipitation-hardenable Ag-containing alloys developed until now, we could find that Ag was mainly added to refine the size of precipitates formed in Mg–RE or Mg–Zn based alloys. To the authors' knowledge, there is no prior research on the development of precipitation-hardenable Mg–Ag based wrought alloys, which motivated us to explore the feasibility of developing precipitation-hardenable alloy based on the Mg–Ag–Ca system.

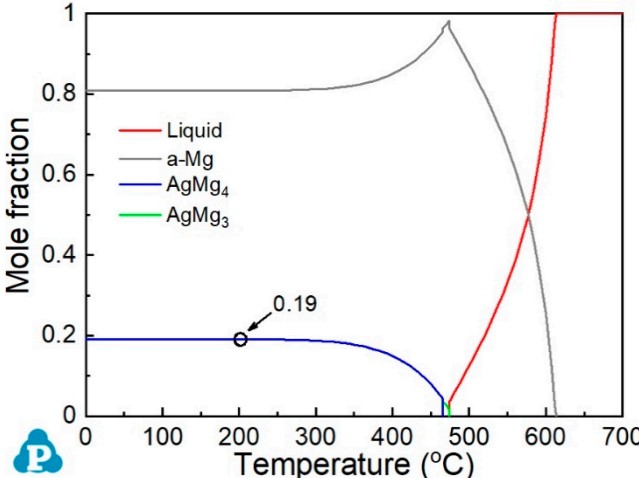

**Figure 1.** Calculated equilibrium mole fraction of phases as a function of temperature for a binary Mg–15Ag (wt.%) alloy using the PANDAT software.

In this study, Mg–xAg–0.1Ca alloys with a composition range from 1.5 wt.% to 12 wt.% were prepared and their age-hardening response at 170 °C was evaluated. Tensile properties of Mg–1.5Ag–0.1Ca, Mg–6Ag–0.1Ca and Mg–12Ag–0.1Ca alloy sheets were evaluated at RT, and a systematic microstructure characterization was conducted to clarify the strengthening mechanism in precipitation-hardenable Mg–Ag–Ca alloys.

## 2. Materials and Methods

Mg–xwt.%Ag–0.1wt.%Ca (x = 1.5, 6 and 12) ingots were produced using pure Mg (>99.9%), pure Ag (>99.9%) and Mg–4.92 wt%Ca master alloy by an induction furnace (IR) under an Argon (Ar) atmosphere. The chemical composition of alloys used in the present study was analyzed by inductively coupled plasma-optical emission spectroscopy (ICP-OES), and the analyzed results were given in Table 1. The as-cast alloys were initially extruded as sheets of 5 mm in thickness at 380 °C. The extrusion ratio and the ram speed were 6 and 5 mm/min, respectively. The extruded sheets were homogenized at 400 °C for 18 h to avoid hot-cracking during rolling, particularly the concentrated Mg–12Ag–0.1Ca alloy. The homogenized sheets were rolled from 5 mm to 1 mm in thickness with ~ 21% thickness reduction per pass by 7 passes. After each pass, the rolled sheets were immediately quenched into cold water. The water quenched sheets were then re-heated to 350 °C prior to subsequent rolling, and rollers were heated to 90 °C during rolling. The rolled sheets were solution-treated at 450 °C for 1 h (T4) and then quenched into cold water. Some of them were subsequently aged in a silicone oil bath at 170 °C.

**Table 1.** Chemical composition of Mg–xAg–0.1Ca (x = 1.5, 6 and 12) alloys analyzed by ICP-OES.

| Alloy | Ag (wt.%) | Ca (wt.%) |
|---|---|---|
| Mg–1.5Ag–0.1Ca | 1.37 | 0.10 |
| Mg–6Ag–0.1Ca | 5.68 | 0.11 |
| Mg–12Ag–0.1Ca | 11.0 | 0.10 |

Dog-bone shaped tensile samples having a parallel length of 12 mm, a width of 4 mm and a thickness of 1 mm were machined from the T4-treated sheets along the RD (0°), 45° and TD (90°). Figure 2 shows a schematic drawing of tensile test samples used in the present study. A screw driven Instron 5565 tensile testing machine was used to evaluate RT tensile properties with a constant testing speed of 2 mm/min. The age hardening response was evaluated by a HM-200 micro Vickers hardness tester (Mitutoyo Corporation, Kawasaki, Japan) under a load of 200 g with a holing time of 10 s. Ten points were measured in each condition. The maximum and minimum hardness values were removed, and the remaining 8-point values were averaged. X-ray diffraction (XRD) patterns were obtained from the mid-layers of sheets using Rigaku RINT Ultima III operating at 40 kV and 40 mA (Rigaku Corporation, Akishima, Japan). Samples for secondary scanning electron microscope (SEM) and electron backscatter diffraction (EBSD) observations were prepared using silicon carbide (SiC) papers, 60 nm alumina suspension, and Ar ion beam using an ELIONIX EIS-200ER ion beam shower system (ELIONIX Inc., Hachioji, Japan). SEM observation was performed at 15 kV using a JEOL JSM-IT500 equipped with a JEOL EX-74600U4L2Q EDS detector (JEOL Ltd., Akishima, Japan). EBSD measurements were performed at 20 kV using the same SEM equipped with TSL OIM 7.0 data collection software (EDAX Inc., Mahwah, NJ, USA). Samples for TEM observation were mechanically ground to about 150 μm in thickness using SiC papers (500, 1000, 2400, and 4000 grit) and subsequently punched to disks of 3 mm in diameter. The disks were twin-jet electropolished at about −50 °C with a solution of 15.9 g lithium chloride, 33.6 g magnesium perchlorate, 1500 mL methanol and 300 mL 2-butoxy-ethanol, and finally ion-milled using a Gatan Precision Ion Polishing System (PIPS) (Gatan Inc., Pleasanton, CA, USA). TEM observations were carried out on JEM-2010 and $C_s$-corrected JEM-ARM200F TEMs operating at 200 kV (JEOL Ltd., Akishima, Japan). The thermodynamic simulation was carried out with the PANDAT software 2020 (CompuTherm LLC, Middleton, WI, USA) [22].

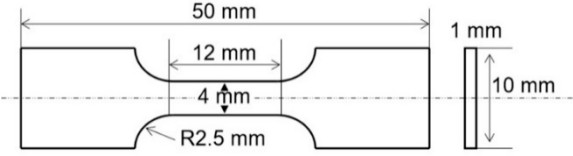

**Figure 2.** Schematic diagram showing the dimensions of tensile test sample used in this study.

## 3. Results

Figure 3 shows the age-hardening response of T4-treated Mg–xAg–0.1Ca alloy sheets (x = 1.5, 6 and 12) during isothermal aging at 170 °C. The 1.5Ag containing alloy sheet shows a low Vickers hardness value of 41.4 HV in the initial condition (T4), and it exhibits a negligible age-hardening. Increasing the Ag content not only leads to a higher hardness value in the T4-treated condition but also enhances the age-hardening response during aging. The 6Ag containing alloy sheet and the 12Ag containing alloy sheet show hardness values of 51.1 HV and 67.0 HV, respectively. After aging for 336 h, the hardness value of the former is increased to 57.4 HV, and that of the latter is increased to 80.0 HV. Table 2 summarizes initial hardness, peak-hardness, time to reach peak hardness and hardness increment of Mg–xAg–0.1Ca alloy sheets.

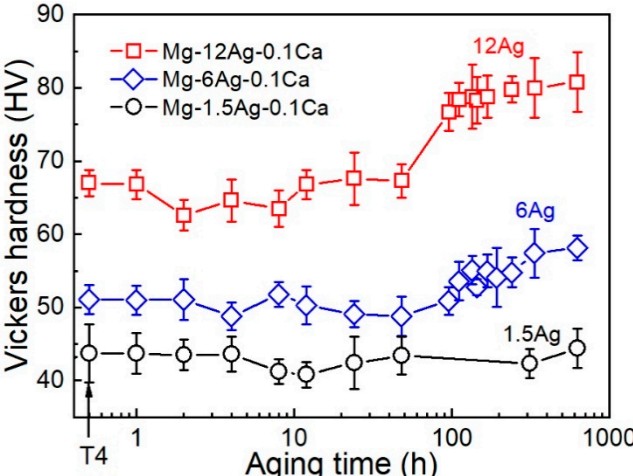

**Figure 3.** Age-hardening response of 1.5Ag, 6Ag and 12Ag containing alloy sheets at 170 °C.

**Table 2.** Initial hardness (T4 condition), peak-hardness, time to reach peak hardness and hardness increment of 1.5Ag, 6Ag and 12Ag containing alloy sheets.

| Alloy | Initial Hardness (HV) | Peak Hardness (HV) | Time to Reach Peak Hardness (h) | Hardness Increment (HV) |
|---|---|---|---|---|
| Mg–1.5Ag–0.1Ca | 41.4 | 43.7 | 0.5 | 2.3 |
| Mg–6Ag–0.1Ca | 51.1 | 57.4 | 336 | 6.3 |
| Mg–12Ag–0.1Ca | 67.0 | 80.0 | 336 | 13.0 |

Figure 4 shows tensile properties of the T4- and T6-treated 6Ag and 12Ag containing alloy sheets at RT. For the purpose of comparison, tensile curves obtained from the T4-treated 1.5Ag containing alloy sheet are given as Figure 4a. The tensile yield strength (TYS) of the T4-treated 1.5Ag containing alloy sheet is measured to be only 85 MPa, 57 MPa and 47 MPa along the RD, 45° and TD, respectively. Increasing the Ag content to 6 wt.% increases the TYS to 112 MPa, 70 MPa and 61 MPa along the RD, 45° and TD, Figure 4b. When the Ag content is further increased to 12 wt.%, the TYS is significantly increased to 193 MPa, 130 MPa and 117 MPa, Figure 4c. Subsequent artificial aging (T6) at 170 °C for 336 h further increases the TYS of 12Ag containing alloy sheet to 236 MPa, 163 MPa and 143 MPa along the RD, 45° and TD, respectively. The uniform elongation (UE) and fracture elongation (FE) of T4-treated 6Ag and 12Ag containing alloy sheets are all higher than 20%. However, both UE and FE are substantially reduced to about 10% after the T6 treatment. The tensile properties of 1.5Ag, 6Ag and 12Ag containing alloy sheets such as TYS, ultimate tensile strength (UTS), UE and FE are summarized in Table 3.

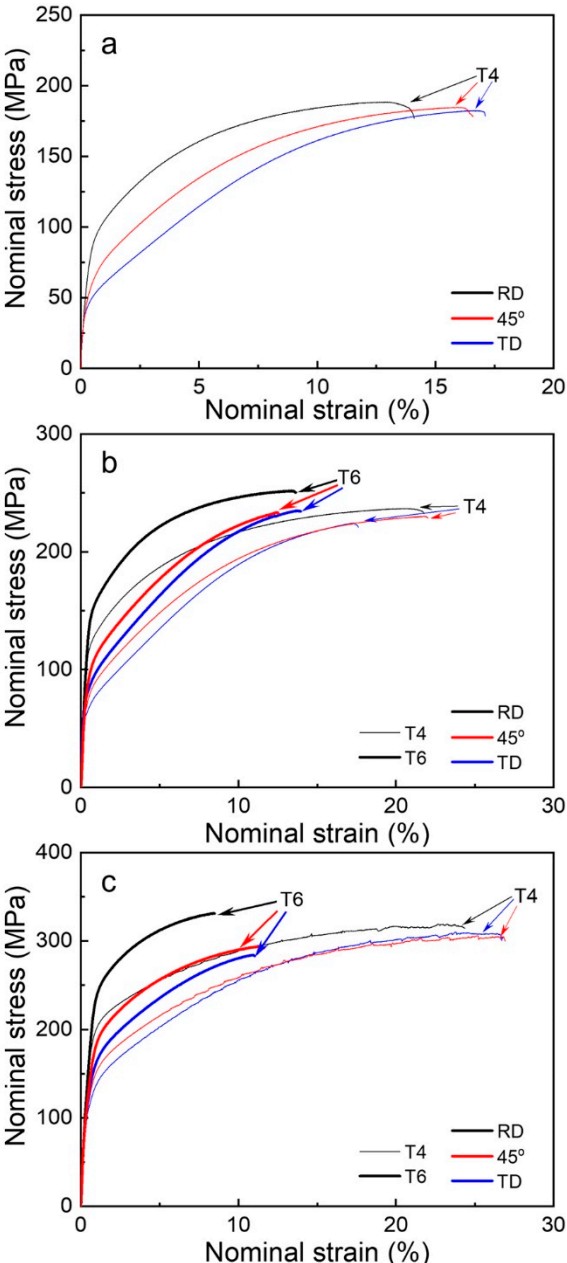

**Figure 4.** Tensile curves obtained from (**a**) T4-treated Mg–1.5Ag–0.1Ca alloy sheet, and T4- and T6-treated (**b**) Mg–6Ag–0.1Ca and (**c**) Mg–12Ag–0.1Ca alloy sheets along RD, 45° and TD at RT.

Figure 5 show EBSD inverse pole figure (IPF) maps and corresponding (0002) and $(10\bar{1}0)$ PFs obtained from the T4-treated 1.5Ag, 6Ag and 12Ag containing alloy sheets. The 1.5Ag containing alloy sheet shows a coarse-grained microstructure of ~277 μm, and abnormal grain growth occurs. Increasing the Ag content to 6 wt.% refines the microstructure and the average grain size is decreased by about one half (~124 μm). In addition, the microstructure homogeneity is enhanced. An increased addition of the Ag content to 12 wt.% causes further refinement in the microstructure. It is to be noted that a homogeneous microstructure consisting of equiaxed grains with an average grain size of ~74 μm is developed in the 12Ag containing alloy. All alloy sheets show a TD-split texture, in which the (0002) basal poles are tilted toward the TD.

**Table 3.** RT tensile properties of 1.5Ag, 6Ag and 12Ag containing alloy sheets stretched along the RD, 45° and TD.

| Alloy | Condition | Direction | TYS (MPa) | UTS (MPa) | UE (%) | FE (%) |
|---|---|---|---|---|---|---|
| Mg–1.5Ag–0.1Ca | T4 | RD | 85 | 189 | 12 | 13 |
| | | 45° | 57 | 185 | 15 | 16 |
| | | TD | 47 | 182 | 16 | 16 |
| Mg–6Ag–0.1Ca | T4 | RD | 112 | 237 | 20 | 21 |
| | | 45° | 70 | 230 | 21 | 22 |
| | | TD | 61 | 224 | 17 | 17 |
| | T6 | RD | 149 | 252 | 13 | 13 |
| | | 45° | 89 | 233 | 12 | 12 |
| | | TD | 81 | 235 | 13 | 14 |
| Mg–12Ag–0.1Ca | T4 | RD | 193 | 319 | 22 | 24 |
| | | 45° | 130 | 306 | 24 | 26 |
| | | TD | 117 | 310 | 23 | 26 |
| | T6 | RD | 236 | 331 | 8 | 8 |
| | | 45° | 163 | 294 | 11 | 11 |
| | | TD | 143 | 284 | 10 | 10 |

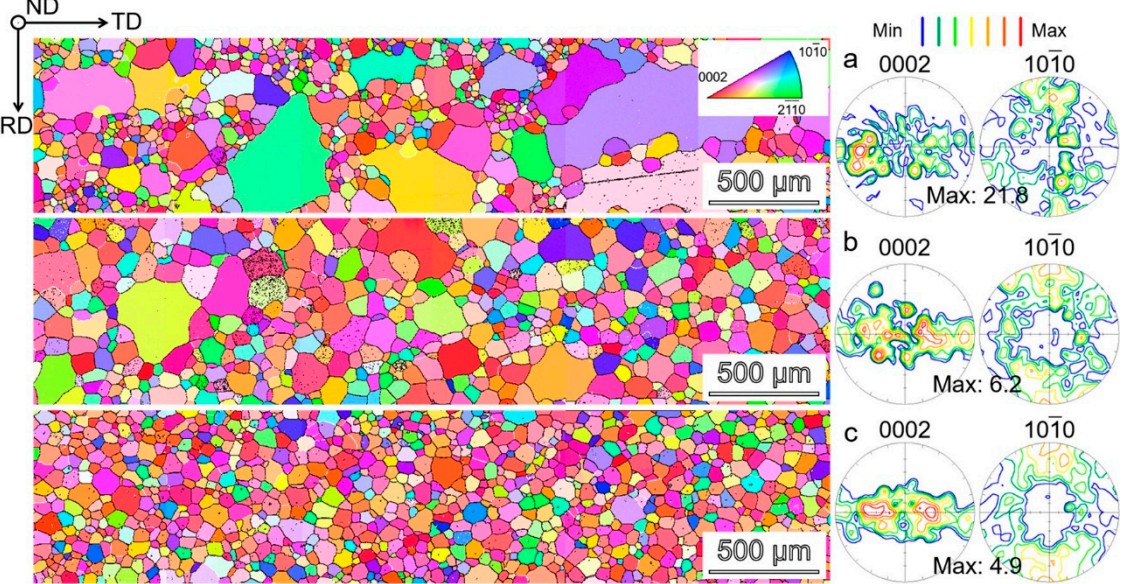

**Figure 5.** EBSD inverse pole figure (IPF) maps and corresponding (0002) and (10$\bar{1}$0) PFs showing microstructures and textures of T4-treated (**a**) Mg–1.5Ag–0.1Ca, (**b**) Mg–6Ag–0.1Ca, and (**c**) Mg–12Ag–0.1Ca alloy sheets.

To understand the reasons why the grain size is decreased with increasing the Ag content, the microstructures of T4-treated samples were observed by SEM. As can be seen from Figure 6a,b, the microstructures of the 1.5Ag and 6Ag containing alloy sheets consist of only α-Mg, indicating that solute atoms are fully dissolved into the Mg matrix after annealing at 450 °C for 1 h. In contrast, the 12Ag containing alloy sheet contains coarse second phase particles that are located both within the grains and along the grain boundaries, Figure 6c. The backscattered electron (BSE) image and corresponding Ag and Ca EDS maps indicate that these particles are mainly enriched with Ag, Figure 6d–f. Point analysis on a particle (marked with a triangle in Figure 6d shows that an atomic ratio between Mg and Ag is close to 4, Figure 6g. Based on our previous study, these particles are believed to be AgMg$_4$ phase [15]. The microstructure of the T4-treated Mg–12Ag–0.1Ca alloy sheet was further observed by TEM. Figure 7a shows the high-angle annular dark-field scanning transmission electron microscopy

(HAADF-STEM) image, in which the contrast is proportional to the square of the atomic number [23,24]. The grain boundary is therefore believed to be enriched with Ag (Z = 47) and or Ca (Z = 20) (Mg, Z = 12) because the grain boundary looks much brighter than adjacent grains. The corresponding Ag and Ca EDS maps reveal that Ag and Ca are co-segregated along the grain boundary, Figure 7b,c.

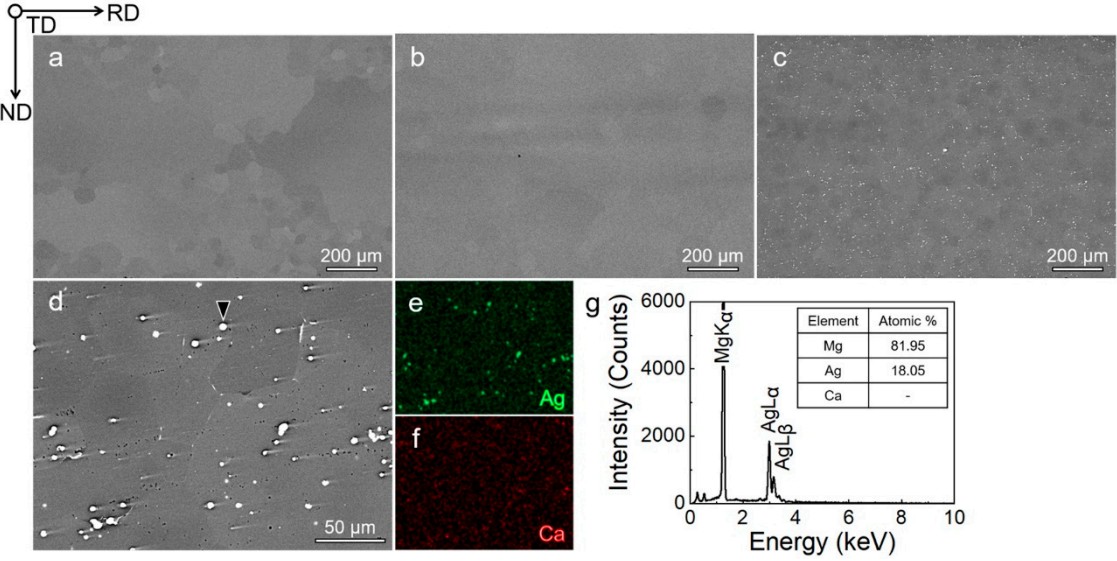

**Figure 6.** Backscattered electron (BSE) images showing microstructures of T4-treated (**a**) Mg–1.5Ag–0.1Ca, (**b**) Mg–6Ag–0.1Ca, and (**c**) Mg–12Ag–0.1Ca alloy sheets. (**d**) BSE image, and corresponding (**e**) Ag and (**f**) Ca elemental maps obtained from the Mg–12Ag–0.1Ca alloy sheet. (**g**) EDS spectrum obtained from a second phase particle (marked with a triangle) in (**d**).

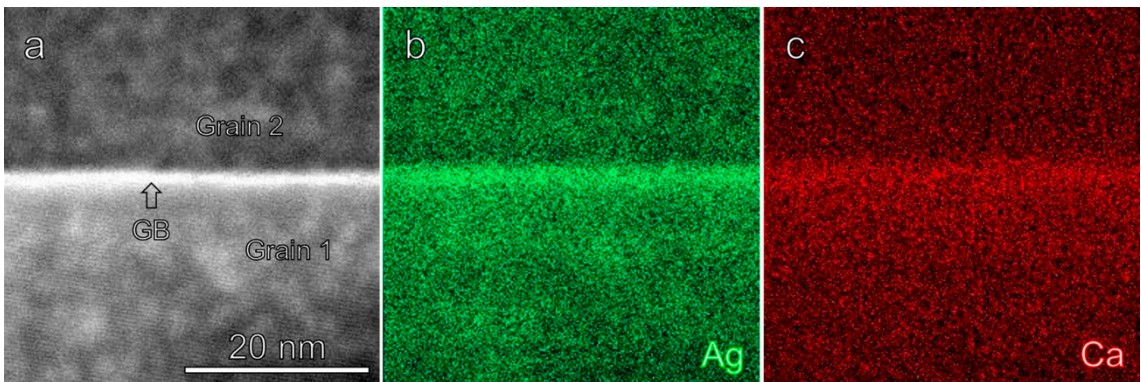

**Figure 7.** (**a**) High-angle annular dark-field-STEM image and (**b**) Ag and (**c**) Ca elemental maps showing the co-segregation of Ag and Ca atoms along a grain boundary of T4-treated Mg–12Ag–0.1Ca alloy sheet.

To clarify the reasons for the enhanced TYS after aging, the microstructure of the T6-treated 12Ag containing alloy sheet was observed by TEM. Figure 8a,b show the bright field (BF) TEM images taken with the incident beam along the $[0001]_a$ and $[10\bar{1}0]_a$ zone axes, respectively. The dispersion of precipitates is homogeneous, and the morphologies of the precipitates seem to consist of rod and polygonal shapes. The rod-type precipitates with a length of 0.5–2.5 µm are on the $\{11\bar{2}0\}_a$ plane and their growing direction is parallel to the $<10\bar{1}0>_a$ or $[0001]_a$ directions. It is to be noted that some rod type precipitates lay on the pyramidal plane, as indicated by triangles in Figure 8b.

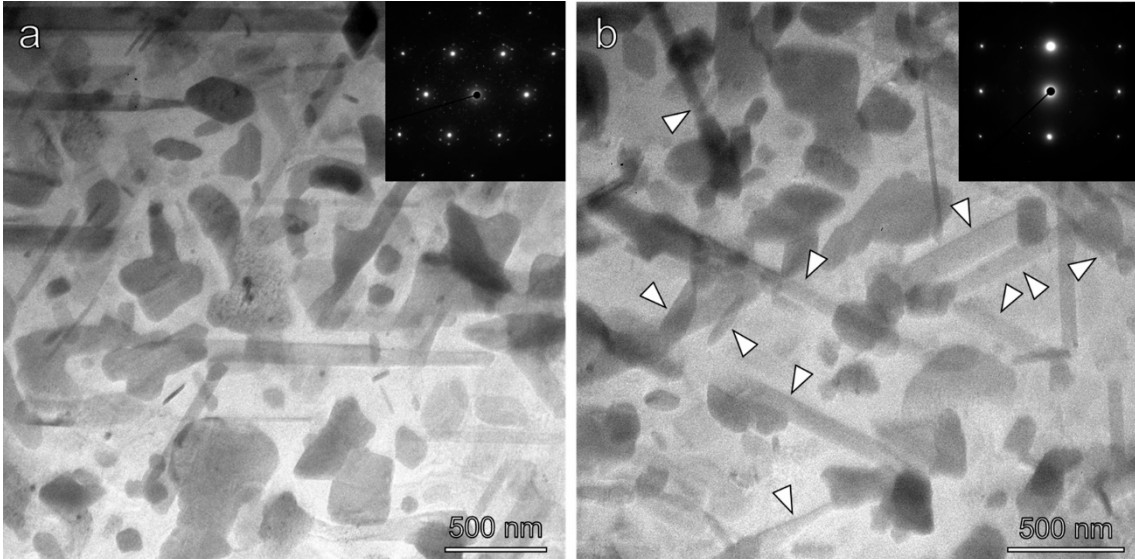

**Figure 8.** Bright field-TEM images showing the microstructure of T6-treated Mg–12Ag–0.1Ca alloy sheet along (**a**) [0001]$_a$ zone axis and (**b**) [10$\bar{1}$0]$_a$ zone axis.

Figure 9a shows the XRD pattern obtained from the T6-treated Mg–12Ag–0.1Ca alloy sheet. For the purpose of comparison, the XRD pattern obtained from the T4-treated sheet is given in Figure 9b. As can be seen that the T4-treated sample mainly consists of $_a$-Mg phase. After the T6-treatment, new peaks are clearly observable, and they are confirmed to be generated mainly by MgAg$_4$ phase. Therefore, the improvement in the tensile yield strength is associated with the formation of AgMg$_4$ precipitates by the T6 treatment.

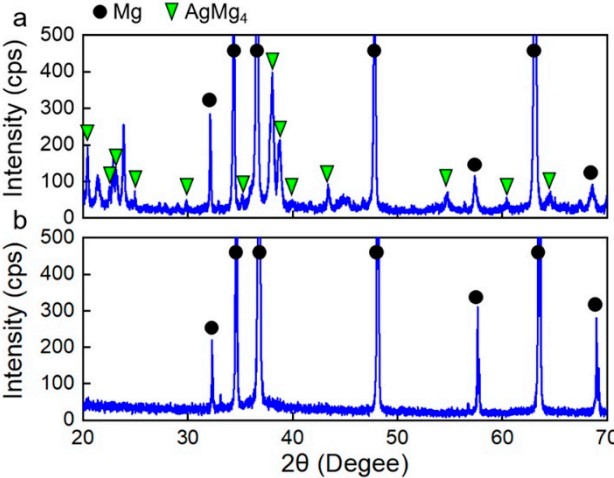

**Figure 9.** (**a**) X-ray diffraction (XRD) pattern obtained from T6-treated 12Ag containing alloy sheet. For the purpose of comparison, XRD pattern obtained from T4-treated 12Ag containing alloy sheet is given in (**b**).

To know whether there is an orientation relationship (OR) between AgMg$_4$ phase and $_a$-Mg matrix, further microstructure analysis was carried out on the T6-treated 12Ag containing alloy sheet. Figure 10a shows the HAADF-STEM image with the zone axis of [0001]$_a$, and Figure 10b shows the high resolution (HR) HAADF-STEM image that is enlarged from the rectangular region in Figure 10a. Figure 10c,d are the fast Fourier transform (FFT) patterns generated from the left-hand region and right-hand region of Figure 10b, respectively. Analysis of these patterns reveals that left-hand region is

the $\alpha$-Mg matrix with the zone axis of $[0001]_\alpha$ and the right-hand region is the $AgMg_4$ phase with the zone axis of $[0001]_{AgMg4}$. By further analyzing the FFT patterns, the OR between the $\alpha$-Mg matrix and $AgMg_4$ is confirmed to be $(0001)_\alpha \parallel (0001)_{AgMg4}$, $[\bar{2}110]_\alpha \parallel [10\bar{1}0]_{AgMg4}$.

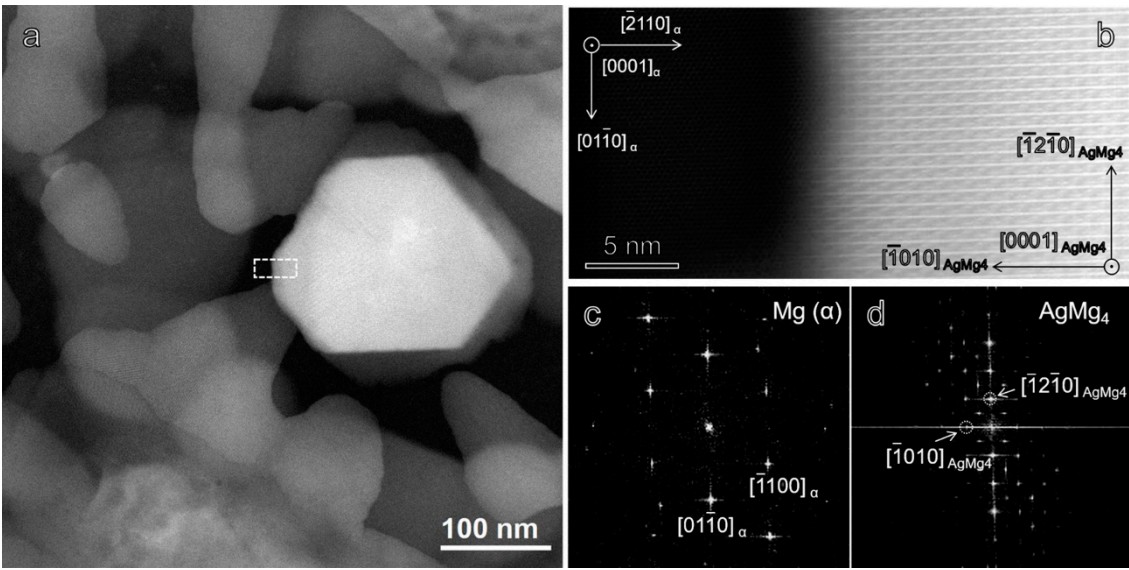

**Figure 10.** (**a**) HAADF-STEM image recorded along the $[0001]_\alpha$ direction. (**b**) Enlargement of the marked region in (a). FFT patterns generated from (**c**) $\alpha$-Mg matrix and (**d**) $AgMg_4$ phase showing their orientation relationship.

## 4. Discussion

In the present study, we have successfully developed a new type of precipitation-hardenable Mg–Ag–Ca sheet alloys. In the T4-treated condition, the 1.5Ag containing alloy sheet shows a large average grain size of ~277 µm. In addition, abnormal grain growth occurs, as shown in Figure 5a. From our previous study, this alloy sheet subjected to 350 °C annealing for 1.5 h exhibits a homogeneous microstructure that consists of equiaxed grains with an average grain size of 20 µm [14], indicating abnormal grain growth is likely to occur at higher temperatures. In the case of the pure Mg and Mg alloy AZ31 (Mg–3Al–1Zn–0.3Mn in wt.%) [25,26], it seems that abnormal grain growth occurs at intermediate temperatures rather than higher temperatures, which is different from the trend observed in the present study. For example, abnormal grain growth occurs at an annealing temperature of 220 °C, while normal grain grow occurs at a higher temperature of 350 °C [25]. It was reported that recrystallized grains with the <11$\bar{2}$0> ∥ RD orientation grow preferentially at the expense of deformed matrix grains close to the <10$\bar{1}$0> ∥ RD orientation and neighboring recrystallized small grains, thereby evolving to abnormally large sizes in pure Mg and AZ31 alloy [27,28]. Figure 11 shows EBSD IPF maps and corresponding IPFs from abnormally coarse grains (>100 µm) and normal-size grains (≤100 µm) in the T4-treated 1.5Ag containing alloy sheet. As can be seen, the normal-size grains have mainly the <11$\bar{2}$0> ∥ RD orientation while the abnormally coarse grains do not have such orientation. These results suggest that the mechanism responsible for abnormal grain growth in Mg–Ag–Ca alloys is likely to be different from that in pure Mg and AZ31 alloy. It is to be noted that a homogeneous microstructure is developed in the 12Ag containing alloy sheet, Figure 5c. Ag and Ca atoms are demonstrated to be enriched in grain boundaries of the T4-treated condition, as shown in Figure 7. It is believed that solute segregation to grain boundaries occurs in the 1.5Ag containing alloy as well. However, the degree of solute segregation should be much lower than that in the 12Ag containing alloy. It is thus hypothesized that the segregated solute atoms effectively reduce grain boundary mobility via solute drag effects, thereby leading to a homogeneous microstructure. This hypothesis is supported by the homogeneous microstructure developed in the 1.5Ag containing alloy sheet after 350 °C annealing for 1.5h. Based on

the PANDAT calculation, the solid solubility of Ag in Mg is substantially decreased from 10 wt.% to 1.42 wt.% with decreasing the temperature from 450 °C to 350 °C, as shown in Figure 12. Considering less Ag atoms will be dissolved into α-Mg matrix and more Ag atoms segregate to grain boundaries at 350 °C, the degree of solute segregation is expected to be much stronger than that at 450 °C. As such grain boundaries are difficult to break away from the solute drag atmosphere and remain pinned in the 1.5Ag containing alloy after 350 °C annealing.

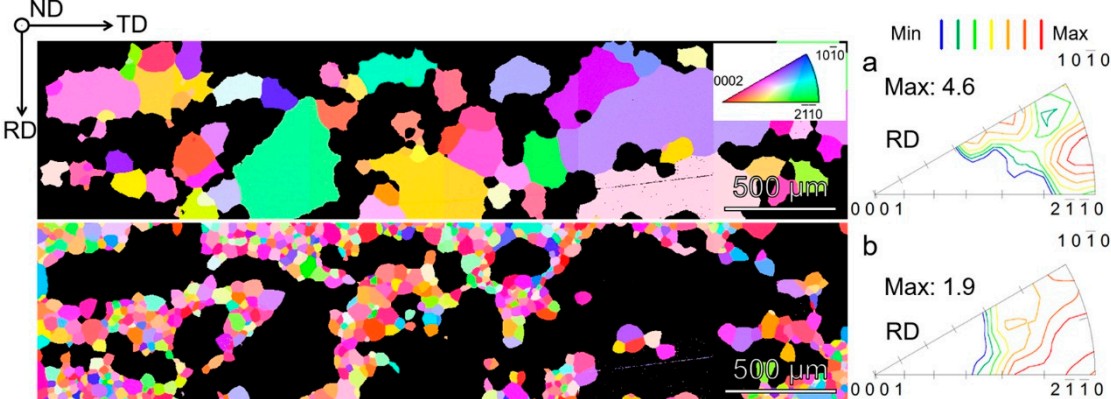

**Figure 11.** EBSD IPF maps and corresponding inverse pole figures showing textures of (**a**) abnormally coarse grains and (**b**) normal-size grains in the T4-treated Mg–1.5Ag–0.1Ca alloy sheet.

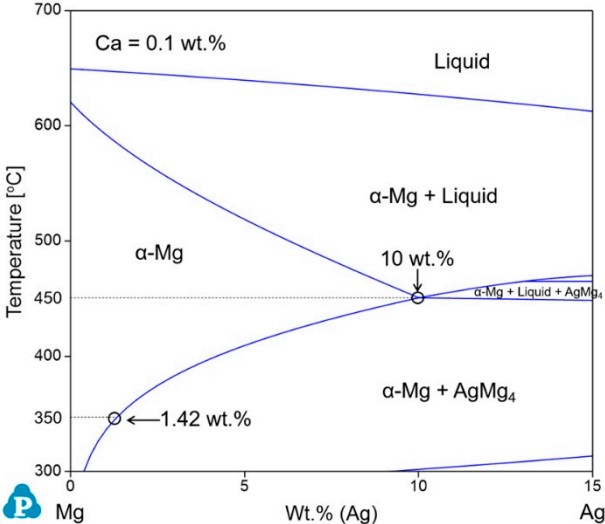

**Figure 12.** Calculated Mg-Ag phase diagram using the PANDAT software, in which the Ca content is fixed as 0.1 wt.%.

In the T4-treated condition, the TYS strength is substantially enhanced by increasing the Ag content from 1.5 wt.% to 12 wt.%. The 12Ag containing alloy sheet shows a homogeneous microstructure with an average grain size of ~74 μm. Thus, the increased TYS can be mainly ascribed to the refinement in the microstructure and solid solution strengthening effect. Nonetheless, the average grain size of the Mg–12Ag–0.1Ca alloy is still much coarser than the those developed in precipitation-hardenable Mg-Zn–Ca–Zr and Mg–Al–Ca–Mn(–Zn) alloys (<10 μm) [8,11,12,29]. This is because fine second phase particles that can effectively retard the growth of recrystallized grains are not present in the 12Ag containing alloy. Addition of small amounts of Al and Mn or Zn and Zr to in the Mg–12Ag–0.1Ca alloy might be promising to reduce the average grain size in the T4-condition by forming a densely distributed nano-scale $Al_8Mn_5$ or $Zn_2Zr_3$ particles [30,31]. On the other hand, all alloys exhibit a higher

TYS along the RD than along the TD due to the formation of the TD-split texture. Such a texture is beneficial for (0001)<11$\bar{2}$0> basal slip and {10$\bar{1}$2} tensile twin to accommodate the plastic deformation along the TD than the RD as their Schmid factor values are higher along the TD than the RD [32–36].

　　The TYS of the 12Ag containing alloy sheet is 193 MPa, 130 MPa and 117 MPa along the RD, 45° and TD, respectively. The T6 treatment further increases the TYS to 236 MPa, 163 MPa and 143 MPa. The strength enhancement by the T6-treatment is rather low compared to other alloys such as Mg–Gd [37], Mg–Zn [38] and Mg–Sn [39] based wrought alloys. Microstructure characterization reveals that non-basal AgMg$_4$ precipitates are formed in the T6-treated sample and their coarse microstructure is the main reason for the inferior age hardenability. There are several ways to refine the size of precipitates by increasing the number density of nucleation sites in early stage of aging: (i) deformation prior to aging treatment [40,41]; (ii) double aging consist of low temperature aging followed by high temperature aging [42–44]; and (iii) addition of trace amounts of elements [21,45,46]. Further optimization of the alloy composition incorporated with the modification of thermomechanical process is therefore expected to improve the age-hardening response and accelerate aging kinetics of Mg-Ag-Ca alloys.

## 5. Conclusions

　　Precipitation-hardenable wrought Mg alloy has been successfully developed based on the Mg–Ag–Ca system. In a T4-treated condition, the TYS of Mg–1.5Ag–0.1Ca alloy sheet is only 85 MPa, 57 MPa and 47 MPa along the RD, 45° and TD, respectively. With increase in the Ag content to 12 wt.%, the TYS is increased to 193 MPa, 130 MPa and 117 MPa along the RD, 45° and TD, which can be mainly ascribed to a refined microstructure and solid solution strengthening effect. Artificial aging at 170 °C for 336 h (T6) further increases the TYS of Mg–12Ag–0.1Ca alloy sheet to 236 MPa, 163 MPa and 143 MPa along the RD, 45° and TD. AgMg$_4$ precipitates lying on the {11$\bar{2}$0}$_\alpha$ and pyramidal planes are responsible for the strength improvement.

**Author Contributions:** Conceptualization, M.B.; methodology, M.B.; validation, X.H. and Y.C.; investigation, M.B. and X.H.; writing—original draft preparation, M.B.; writing—review and editing, X.H. and Y.C.; project administration, Y.C.; funding acquisition, M.B. and X.H. All authors have read and agreed to the published version of the manuscript.

**Funding:** This research was funded by JSPS KAKENHI, grant numbers JP20K15067 and JP18K04787.

**Conflicts of Interest:** The authors declare no conflict of interest. The funders had no role in the design of the study; in the collection, analyses, or interpretation of data; in the writing of the manuscript, or in the decision to publish the results.

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
