# Peer review of "Microstructures and Mechanical Properties of Precipitation-Hardenable Magnesium–Silver–Calcium Alloy Sheets"

_metals, doi:10.3390/met10121632_

Round 1
Reviewer 1 Report
Sentence formulation could be in occasional cases improved (subjective view), hovewer it is fully understandable.
- Authors should unite terms: sample or specimen, not both of them.
- Missing the information if the hardness value is average or not (at least 3 to 5 intents on every sample would be apropriate).
- Auhotrs using a word "abnormal" but did not specify what is considered to be normal size. I mean heterogenity is obvious and I understood what authors are saying, but if there is anything abnormal, normal definition should be provided.
- Line 95 description of sample dimmensions for tensile strength in not enough described. make a more accurate description or attach a figure with dimmensions.
- Line 157 Markers on figure 4 are barely visible, i would suggest to remove the contour and remain just plain white or black text, whatever suits better and improves readability.
Reviewer 2 Report
Interesting work on the assessment of the microstructure and properties of magnesium alloys with the addition of Ag and Ca after heat treatment. The plan and scope of research proposed in the work are correct. The methodology is also appropriate to achieve the goal of assessing and characterizing the structure and properties of the tested alloys. The discussion of the results fully justifies the obtained research results.
Author Response
We appreciate the reviewer for acknowledging the importance of our work.
Reviewer 3 Report
This manuscript fits into the topic of precipitate hardenable Mg alloys not alloyed by rare earth elements. In this work Mg alloys containing 0.1 wt. % Ca and Ag in amounts varying from 1.5 to 11 wt. % were cast, extruded, rolled and thermally treated (T4, T6). The produced materials were subjected to mechanical tests (hardness and tensile testing) and their microstructure was studied by microscopy.
The main result of this study is a hardening efect found with increasing amount of Ag in the alloy and ascribed to grain refinement and precipitates. The precipitates were identified as AgMg4 lying in {11-20} and pyramidal planes. It is the opinion of the reviewer that it is also an effect of solution hardening but this is not mentioned in the manuscript.
The discussion of the results is a bit confusing because the authors state that the effect of the thermal treatment applied in this study is rather low by comparison to the literature evidence on other types of Mg alloys (cit. "their coarse microstructure is the main reason for the inferior age hardenability). In what follows, it is claimed, basing on the literature results, what should be done for the optimization of the alloy properties.
In the Conclusions, however, it is stated that the above mentioned precipitates are responsible for the strength improvement.
The reviewer therefore recommends to clarify the text in these sections, first summarizing the achieved results, then saying whether the authors are happy with these results and finally to propose further steps to the alloy optimization. This text should be coherent with the Conclusions.
In the opinion of the reviewer, the authors present results of an original study aiming at the improvement of mechanical properties of Mg alloys, which is up-to-date and interesting for the reader. the manuscript, however, should be revisited to make the text clearer to the reader.
Minor correction: In the second phrase of Conclusions, please, add "TD" before "respectively".
In conclusion, the manuscript can be accepted after minor revision.
